# Evaluation of Bacterial Diversity and Evolutionary Dynamics of Gut *Bifidobacterium longum* Isolates Obtained from Older Individuals in Hubei Province, China

Zhendong Zhang,[a,b] Qiangchuan Hou,[a,b] Yurong Wang,[a,b] Fanshu Xiang,[a,b] Zhuang Guo[a,b]

[a]Hubei Provincial Engineering and Technology Research Center for Food Ingredients, Hubei University of Arts and Science, Xiangyang, Hubei, People's Republic of China
[b]Xiangyang Lactic Acid Bacteria Biotechnology and Engineering Key Laboratory, Hubei University of Arts and Science, Xiangyang, Hubei, People's Republic of China

**ABSTRACT** *Bifidobacterium longum* predominates in the human gut throughout the life span, from birth to old age, and could alter the intestinal microbial population and immune function in the elderly. We investigated the intestinal bacterial diversity in the elderly, and further evaluated the genetic diversity and population structure of *B. longum*. The results revealed a distinct difference in gut bacterial populations between the elderly from Xiangyang and its neighboring region, Enshi city. A total of 62 bifidobacterial strains were isolated, 30 of which were found to be *B. longum*. The multilocus sequence typing (MLST) analysis also revealed that 437 *B. longum* isolates from diverse regions worldwide, including the 30 isolated in this study, could be classified into 341 sequence types (STs). They could be further clustered into 10 clonal complexes and 127 singleton STs, indicating a highly genetic diversity among *B. longum* isolates. Two putative clone complexes (CCs) containing the isolates from Xiangyang were found to be geographically specific, and a 213-bp recombination fragment was detected. Phylogenetic trees divided these 437 isolates into three lineages, corresponding to the three subspecies of *B. longum*. It is noteworthy that two isolates from the elderly were identified to be *B. longum* subsp. suis, while the others were *B. longum* subsp. longum. Together, our study characterized the intestinal bacterial diversity and evolution of *B. longum* in the elderly, and it could contribute to further studies on the genotyping and discrimination of *B. longum*.

**IMPORTANCE** *Bifidobacterium longum* are common inhabitants of the human gut throughout the life span, and have been associated with health-promoting effects, yet little is known about the genotype profile and evolution of these isolates. Our study showed that there was significant difference in gut bacterial community and abundance of *B. longum* between the elderly from two neighboring cities. Furthermore, the possible geographically specific STs, CCs, and intraspecies recombination fragment were found among the *B. longum* isolates from elderly.

**KEYWORDS** elderly, gut microbiome, *Bifidobacterium longum*, MLST, genotyping

Address correspondence to Zhuang Guo, guozhuang@vip.163.com.
The authors declare no conflict of interest.

The gut microbiota constitutes a complex ecological community. There are trillions of microbes inhabiting the human intestinal tract, and the most abundant genera in the human gut are *Lactobacillus*, *Propionibacterium*, *Streptococcus*, *Bacteroides*, *Corynebacterium*, *Staphylococcus*, *Moraxella*, *Haemophilus*, *Prevotella*, and *Veillonella* (1). This complex ecological community influences both the normal physiology and disease susceptibilities of the host. In turn, intestinal microbial diversity is influenced by factors such as diet, genetics, age, environment, and antibiotic use (2). Aging is one of the most important factors affecting intestinal microbial diversity. During aging, there are changes in several factors such as physiology, diet, and lifestyle, and these lead to alterations in the intestinal microbial diversity. These alterations include a decrease in the diversity of the gut microbiome (3, 4), an increase in Proteobacteria (5), and a higher relative abundance of *Bifidobacterium* in younger

individuals and lower abundance in the elderly. Further, the highest relative abundance of the genera *Prevotella* and *Bacteroides* is observed in the 19- to 24-year age group (2). Together, these age-related changes can affect human health.

Bifidobacteria are common inhabitants of the human gut along with lactobacilli, which are the most common species of probiotics. Of all bifidobacterial species, *Bifidobacterium longum* shows a high prevalence in the gut throughout the life span in humans, from birth to old age (6, 7), and it can even be isolated from centenarians (8). *B. longum* could have several health-promoting effects in hosts, including protection against autoimmune diseases, metabolic syndromes such as obesity and hyperlipidemia, and brain–gut disorders (9). Interestingly, accumulating evidence shows that *B. longum* may modulate the intestinal microbial population and immune function in the elderly (10, 11).

*B. longum* isolates were first discovered and isolated by Reuter in 1963. In 2008, Mattarelli et al. demonstrated that there were three subspecies of *B. longum*: *Bifidobacterium longum* subsp. longum, *Bifidobacterium longum* subsp. infantis, and *Bifidobacterium longum* subsp. suis (12). In 2015, a fourth subspecies—*B. longum* subsp. suillum—was identified from the feces of piglets using multilocus sequence typing (MLST) and amplified fragment length polymorphism (AFLP) analyses (13). However, Albert et al. argued against its designation as the fourth novel subspecies of *B. longum*, explaining that this subtype did not meet the relevant criteria (14).

Nevertheless, it is known that these closely related species and subspecies have similar nutritional requirements. Although there is heterogeneity among *B. longum* isolates (15), it is difficult to precisely discriminate them using morphological and physiological characteristics. However, probiotic effects, fermentation characteristics, antimicrobial susceptibilities, and sensorial attributes are strain-specific, making it vital to distinguish and select specific strains for developing probiotics or starter cultures (16).

Genotyping—including AFLP (17, 18), random amplified polymorphic DNA (15, 19), pulsed-field gel electrophoresis (15), multilocus variable number of tandem repeat analysis (20), and MLST analyses (13, 17, 18, 21, 22)—has been used to examine *B. longum* isolates. Of these, MLST was first used in 1998 for the evaluation of *Neisseria meningitidis* (23) and then started being used widely to evaluate epidemiological characteristics and for the typing analysis of bifidobacteria and lactobacilli. MLST can be used to accurately discriminate different isolates within a species and subspecies by comparing allelic profiles, providing precise information on strain evolution, and performing population analysis and classification (17, 18, 21, 22, 24). It has a high resolution and can provide unambiguous genotype nomenclature and information that can be easily shared between laboratories (25). Previous studies have shown that MLST can be used for the subspeciation (13) and species delineation (22) of *B. longum* and to track the transmission of intestinal *B. longum* (18, 21).

Despite the potential effect of *B. longum* on health, the genetic diversity of *B. longum* isolates in the guts of the elderly has never been investigated. In addition, the number of isolates evaluated previously has been rather limited. To this end, this study aimed to examine the bacterial population and diversity in the guts of elderly individuals from Hubei Province, China, and to further evaluate and compare the genotypic diversity of gut *B. longum* isolates from the elderly in this region to that of isolates from different sources using MLST analysis.

## RESULTS AND DISCUSSION

**Bacterial diversity in the guts of the elderly.** Using 16S rRNA gene amplicon sequencing (Table S1), 2,093,929 high-quality bacterial 16S rRNA V3-V4 region reads were obtained from the 42 stool samples, and each sample contained 36,879 to 67,654 (49,855 ± 7,329) reads. We detected 320,852 OTUs (cutoff, 97% sequence similarity), and each sample had 7,639 OTUs on average (range: 1,663–13,484; SD = 2,688). The OTU numbers for each stool sample from Xiangyang city individuals was 9133.35 (SD = 2,211.17) and that for those from Enshi city was 6281.14 (SD = 2,371.49). In addition, the rarefaction curve based on the observed species and Shannon indices suggested that all samples produced sufficient data for further analysis (Fig. S1).

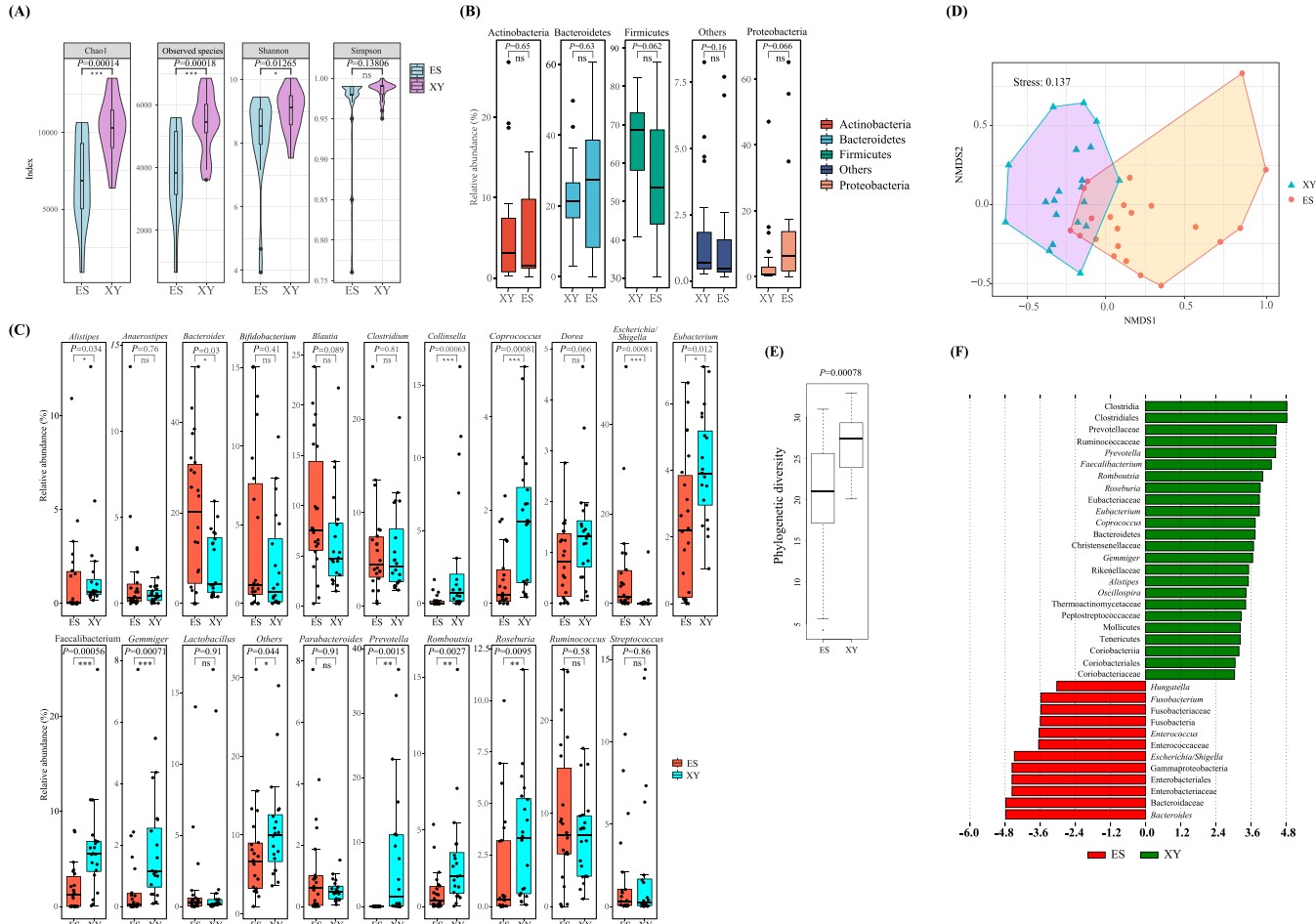

**FIG 1** Evolution of $\alpha$-diversity and $\beta$-diversity in stool samples from the elderly residing in Xiangyang city (XY) and Enshi city (ES). (A) Violin plots of $\alpha$-diversity indices (observed species, Shannon, and Simpson indices) of fecal samples. (B) Boxplot of dominant phyla observed in the stool samples from the elderly residing in different regions. (C) Dominant genera in stool samples from the elderly residing in different regions. Bacterial taxa from (B) and (C) with a mean relative abundance of ≥1.0% were defined as "dominant" and those with a mean relative abundance of <1.0% as "others." (D) Non-metric multidimensional scaling plot of bacterial community structure based on Bray-Curtis distance. (E) Phylogenetic diversity analysis of intestinal bacterial community of the elderly between the Enshi and Xiangyang city. (F) LEfSe analysis of relative bacterial abundance in stool samples. The linear discriminant analysis score (threshold, 3.0) identified the size of differentiation between groups. All *P* values were calculated using the Kruskal–Wallis test (ns, *P* > 0.05; *, *P* < 0.05; **, *P* < 0.01; ***, *P* < 0.001).

The species richness and evenness of the stool samples are presented in Fig. 1A. Species richness indices (observed species index and Chao1 index) were significantly higher in the elderly from Xiangyang city than in those from Enshi city (*P* < 0.05). The Shannon index of the elderly from Xiangyang city was significantly higher than that of individuals from Enshi city; however, the Simpson index showed no significant difference between the two groups. Previous studies have shown that microbial diversity is a measure of microbiome health, and decreased microbial diversity is correlated with increased frailty, inflammation, and disease prevalence (3, 4). The greater gut microbiome diversity observed in the elderly from Xiangyang city indicates healthier intestinal flora in these individuals.

**Bacterial community analysis.** An average of 98.46% of reads could be assigned to four dominant phyla (relative abundance ≥1.0%) (Fig. 1B). The four dominant phyla—Firmicutes, Bacteroidetes, Actinobacteria, and Proteobacteria—showed no significant difference in terms of relative abundance between the elderly from Xiangyang city and Enshi city. Firmicutes were the most abundant, followed by Bacteroidetes, Actinobacteria, and Proteobacteria. These four phyla were also found to be the most predominant in previous studies (26, 27).

The bacterial profile at the genus level was also evaluated. The elderly from Enshi city showed 14 dominant genera—*Bacteroides*, 19.76%; *Ruminococcus*, 9.63%; *Blautia*, 9.44%;

*Escherichia/Shigella*, 6.91%; *Clostridium*, 6.18%; *Bifidobacterium*, 4.26%; *Streptococcus*, 1.67%; *Eubacterium*, 2.41%; *Faecalibacterium*, 1.93%; *Roseburia*, 1.81%; *Anaerostipes*, 1.51%; *Lactobacillus*, 1.33%; *Alistipes*, 1.31%; and *Parabacteroides*, 1.17%. In contrast, there were 17 predominant genera in the elderly from Xiangyang city—*Bacteroides*, 8.70%; *Prevotella*, 7.77%; *Ruminococcus*, 7.47%; *Blautia*, 6.55%; *Faecalibacterium*, 6.14%; *Clostridium*, 6.01%; *Eubacterium*, 4.08%; *Roseburia*, 3.30%; *Romboutsia*, 3.01%; *Collinsella*, 2.82%; *Streptococcus*, 2.45%; *Bifidobacterium*, 2.44%; *Lactobacillus*, 1.76%; *Coprococcus*, 1.76%; *Gemmiger*, 1.72%; *Alistipes*, 1.67%; and *Dorea*, 1.39%. Altogether, these 20 genera accounted for an average of 71.84% of the total sequences (see Fig. 1C), and 13 of them belonged to the phylum Firmicutes. Of these 13, 12 (i.e., *Bacteroides, Prevotella, Ruminococcus, Blautia, Faecalibacterium, Clostridium, Eubacterium, Roseburia, Streptococcus, Bifidobacterium, Lactobacillus,* and *Alistipes*) were predominant in both groups. Moreover, 11 of the 20 dominant genera showed significant differences in relative abundance between the two groups: The abundance of *Alistipes, Escherichia/Shigella,* and *Bacteroides* was higher in the elderly from Enshi city, whereas that of *Coprococcus, Prevotella, Faecalibacterium, Eubacterium, Roseburia, Romboutsia, Collinsella,* and *Gemmiger* was higher for the elderly from Xiangyang city (Fig. 1C). These results indicate the differences in the microbiota between the two groups of the elderly. They were further supported by a non-metric multidimensional scaling (NMDS) plot based on weighted Bray-Curtis distance (stress <0.2), which showed that the bacterial communities in the elderly from the two regions in Hubei Province tended to form two distinct groups (Fig. 1D). Furthermore, the phylogenetic diversity analysis revealed a significant difference in the intestinal bacterial communities between the two regions ($P < 0.001$; Fig. 1E).

Using LEfSe analysis, 22 taxa were identified (Fig. 1F). Two phyla, three classes, two orders, eight families, and nine genera were more common in the elderly from Xiangyang city, whereas one phylum, one class, one order, four families, and five genera were more highly enriched in the elderly from Enshi City. The most differentially predominant bacterial genera for the elderly from Xiangyang city were *Faecalibacterium, Prevotella,* and *Romboutsia,* and for the elderly from Enshi, these were *Escherichia/Shigella* (Enterobacteriaceae) and *Bacteroides* (LDA score [$\log_{10}$] > 4). These findings were consistent with results of the bacterial community structure analysis.

The intestinal microbiome is an important determinant of human health, and its diversity can be influenced by diet, lifestyle, diseases, and medication use (28–30). In the current study, although the microbial phylum profiles were similar, the species community richness, bacterial community structure analysis, and NMDS plot showed great genus-level differences in bacterial profiles between the guts of the elderly from Xiangyang city and Enshi city. Enshi city is located in a mountainous region with a higher altitude, and its population includes a higher number of Tujia and Miao people. In contrast, Xiangyang city has a lower altitude, and the residents are Han people. Therefore, compared with Xiangyang city, Enshi city has more diverse fermented and pickled vegetables with more pepper, including the typical traditional fermented food, Zha-chili, owing to differences in the lifestyles of its population. In addition, differences in the selenium content of crops and vegetables also contribute to differences in diet between the two regions (31, 32). We suggested that diet-related differences may be the key factor responsible for the differences in intestinal bacterial community structures between the two regions (33).

A higher $\alpha$-diversity is thought to be related to a high consumption of fermented foods with high salt content (27, 34). Previous studies have suggested that an intestinal flora with the highest relative proportion of *Bacteroides* and lowest proportion of *Prevotella* is related to the consumption of a western diet that is high in animal protein, choline, and saturated fat content (35). *Prevotella* abundance increases with carbohydrate intake, as its enzymes can degrade plant fibers and contribute to polysaccharide breakdown (36, 37). In the current study, the intestinal flora with a higher relative proportion of *Bacteroides* and lower proportion of *Prevotella* was observed in the elderly from Enshi city. These individuals also showed higher predominance of *Escherichia/Shigella* spp., which are usually thought to be harmful and certain strains of which are capable of causing diarrheal disease. Castaño-Rodríguez et al. showed that intestinal

flora with a predominance of *Escherichia/Shigella* spp. is usually observed in the guts of patients with a particular disease, and these people tended to more often be in a proinflammatory state (38).

*Prevotella*, *Faecalibacterium*, *Eubacterium*, and *Roseburia* were four of the 11 genera that showed significant differences in relative abundance between the elderly from Xiangyang city and Enshi city. The four genera were enriched in the elderly from Xiangyang. These genera belong to the phylum Firmicutes, and specifically to clostridial clusters IV and XIVa, which are reported to be butyrate-producers (39). Studies have shown that microbial butyrate increases gut barrier function (40, 41). Hence, these Gram-positive, strictly anaerobic, saccharolytic bacteria are speculated to be beneficial for maintaining gut barrier function. They also have immunomodulatory and anti-inflammatory properties, associated with a higher quality of life and improved mental health (39, 42).

*Bifidobacterium* spp. can colonize the gut (43); repress harmful enzymatic activities within the microbiota (44); activate a number of dietary compounds to generate healthy bioactive molecules (45); produce vitamins (46, 47), GABA (48), conjugated linoleic acids (49), and short-chain fatty acids (50); and also reduce inflammation in the host (51, 52). Despite the differences in daily diet, the relative abundance of *Bifidobacterium* was not significantly different between the gut of the elderly from Enshi and Xiangyang city. Furthermore, there have been very few studies on culturable bifidobacteria in the guts of elderly individuals. Therefore, we isolated and identified bifidobacteria isolates from the 42 stool samples.

**Isolation and identification of bifidobacteria.** A total of 62 possible bifidobacterial strains were obtained from the 42 stool samples, including 50 (accounting for 81%) from the elderly in Xiangyang city and only 12 strains (19%) from those in Enshi city (Fig. 2; Table S2). These strains were all Gram-positive, catalase-negative, and F6PPK-positive. Table S2 provides the pairwise similarities in the 16S rRNA gene sequences of the 62 bifidobacterial isolates, consistent with the results of phylogenetic analysis. Among the 62 isolates, 1, 6, 6, 2, 30, and 17 isolates showed sequence similarities of over 99% to *B. adolescentis*, *B. dentium*, *B. bifidum*, *B. faecale*, *B. longum*, and *B. pseudocatenulatum* strains and were classified as these species, respectively. Isolates of *B. longum* and *B. pseudocatenulatum* were the most common, accounting for 48.39% and 27.42% of the isolates, respectively.

*B. longum* is widely reported to play a role in host health and disease susceptibility (51–53). *B. longum* is among the first species to colonize the human gut and is present in the gut microbiota throughout an individual's life span (6, 7). The abundance of *B. longum* varies with age (9). Turroni et al. showed that in the guts of infants and adults, culturable *B. longum* isolates account for the highest proportion of all bifidobacteria (54). Similarly, in the current study, *B. longum* isolates accounted for the highest proportion in the guts of the elderly, too.

Using high-throughput sequencing, we found that there was no difference in the relative abundance of *Bifidobacterium* in the guts of the elderly between Xiangyang city and Enshi city. However, the number of isolates obtained differed greatly. It is noteworthy that only three *B. longum* isolates were recovered from three of the 20 elderly individuals in this study; while 27 *B. longum* isolates were recovered from 19 of the other 22. The rate of *B. longum* isolation was lower in the elderly from Enshi city than in those from Xiangyang city. Using microbial cultures, Martín et al. showed that *B. longum* isolates can be recovered from centenarians. However, many researchers have shown that there is a lower proportion of *B. longum* in the guts of the elderly (6, 55). We speculated that apart from ageing, dietary factors may have also led to a low relative abundance of *B. longum* in the intestinal tract of the elderly from Enshi city. Considering their reported benefits to human health, *B. longum* strains isolated from the elderly were selected for further analysis using MLST.

**Genetic diversity of the MLST loci in *B. longum*.** A total of 437 *B. longum* isolates were included in the MLST analysis. These consisted of 30 isolates from our study and 407 previously identified *B. longum* genomes average nucleotide identity [ANI] values for these 407 genomes when compared against the type strain *B. longum* subsp. longum DSM 20219 were all greater than 95%, see Fig. S2a (56). The GC content and genomic length were 59.97% and 2.41 Mb (Fig. S2b and S2c), respectively.

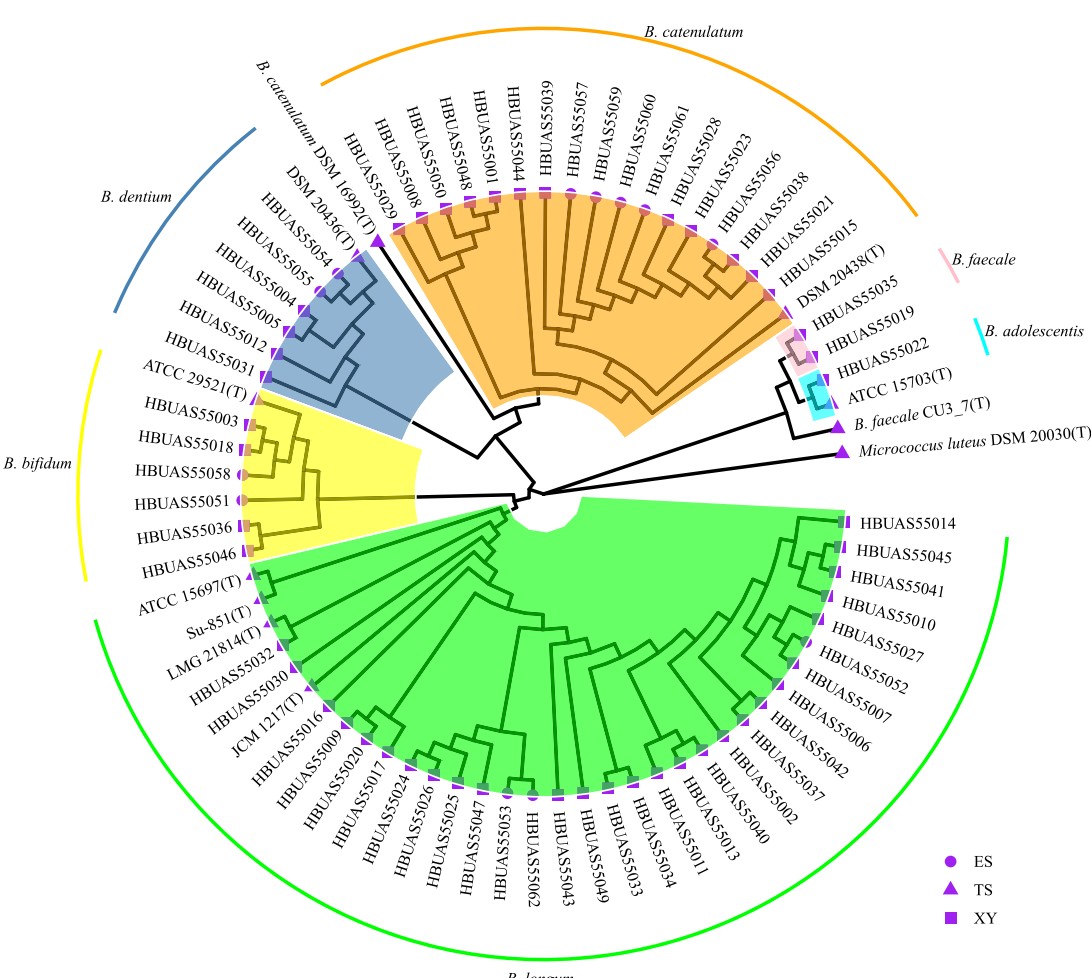

**FIG 2** Neighbor-joining phylogenetic tree based on the 16S rRNA sequences of the 62 bifidobacterial gut isolates from the elderly residing in Xiangyang city and Enshi city. MEGA X software and the Kimura 2 parameter model were used for generating the phylogenetic tree. The number of bootstrap replicates was set at 1,000. The tree was visualized using the R package ggtree. Circles (ES) and squares (XY) at the tips indicate that the isolates were from the elderly in Enshi city and Xiangyang city, respectively; triangles (TS) indicate that the isolates were type strains downloaded from the NCBI database.

For all 437 isolates, the nucleotide diversity within each partial sequence of the six housekeeping genes (*clpC*, *fusA*, *ileS*, *purF*, *rplB*, and *rpoB*) was determined (Table 1). The number of alleles per locus ranged from 43 to 88, and the number of polymorphic sites per housekeeping gene varied from 59 (*rplB*) to 116 (*clpC* and *purF*). In total, 545 polymorphic sites were detected for the 437 *B. longum* isolates, and together, these isolates formed 341 STs. The GC content of the six partial housekeeping genes ranged from 60.1 (*fusA*) to 65.9%

**TABLE 1** Descriptive of genetic variability in six loci of 437 *B. longum* isolates

| Locus | Length (bp) | No. of Alleles | Polymorphic sites | GC content (% mol) | Tajima's D | Phi-test | $\pi^a$ (per site) | dN/dS[b] |
|---|---|---|---|---|---|---|---|---|
| *clpC* | 600 | 88 | 116 | 0.626 | −1.82707*[c] | 0.27943* | 0.01195 | 0.03677 |
| *fusA* | 664 | 86 | 88 | 0.601 | −1.17878 | 0.428956 | 0.01254 | 0.02672 |
| *ileS* | 487 | 75 | 94 | 0.624 | −1.30697 | 0.220715 | 0.01708 | 0.01692 |
| *purF* | 574 | 58 | 116 | 0.613 | −1.18919 | 0.289855 | 0.02023 | 0.01768 |
| *rplB* | 355 | 43 | 59 | 0.659 | −2.45117** | 0.208791** | 0.01315 | 0.1405 |
| *rpoB* | 497 | 88 | 72 | 0.644 | −0.8928 | 0.362821 | 0.02158 | 0.01934 |
| Concatenated | 3177 | 341 | 545 | 0.625 | −1.55697 | 0.354986*** | 0.01355 | 0.02392 |

[a]$\pi$, the mean pairwise nucleotide differences per site.
[b]dN/dS, the ratio of non-synonymous (dN) and synonymous (dS) substitutions of selective pressure on each locus.
[c]*, $P < 0.05$; **, $P < 0.01$; ***, $P < 0.001$.

(*rplB*), consistent with the GC content of the *B. longum* genome. The nucleotide diversity ($\pi$) at the six loci varied between 0.01195 (*clpC*) and 0.02023 (*purF*). Tajima's D test showed negative values for all six housekeeping genes. The *dN/dS* ratio for each locus varied between 0.01692 and 0.1405, and that for the concatenated sequences was 0.02392, indicating that these genes had undergone negative selection. Previously, Delétoile showed that the *rpoB* gene shows the highest *dN/dS*, and that *purF* has the most polymorphic sites (21, 22). These findings were consistent with our results.

**STs and clonal complexes in *B. longum*.** Allele numbers were assigned for each unique allele in the 437 isolates. Based on allele profiles, an ST number was assigned. Overall, a total of 341 STs were observed (Table S3). Of these STs, 44 only corresponded to a single isolate, and 56 included at least three strains. ST29 formed the largest group and was composed of 14 isolates, followed by ST24 (six isolates) and ST101 and ST21 (each with five strains). There were 52 STs composed of two to four isolates. Furthermore, the four type strains employed in this study—*B. longum* subsp. infantis DSM 20088, *B. longum* subsp. suis DSM 20211, *B. longum* subsp. longum DSM 20219, and *B. longum* subsp. suillum Su-851 —were assigned to ST29 (containing the most isolates), ST50 (containing only one isolate), ST34 (containing four isolates), and ST25 (containing only one isolate), respectively.

The 341 STs could be clustered into 10 clone complexes (CCs) and 127 singleton STs (Table S3). Of the identified CCs, CC1 was the most common, accounting for about 65% of all isolates, followed by CC2 (composed of nine isolates) and CC8 (composed of three isolates). In contrast, CC3 to CC7 and CC9 contained two isolates each.

The relatedness between *B. longum* STs was also determined using PHYLOViZ (57): two major BURST groups (BGs; consisting of three or more STs), eight minor BGs (consisting of two STs), and 93 singleton STs were identified (Fig. 3B; Table S3). Analyzing concatenated sequences of the six loci, we found that the largest BG (BG1) contained 94 STs distributed throughout the phylogenetic tree. The network profile among the *B. longum* STs obtained using goeBURST was consistent with the relationship observed using Prim's algorithm (BioNumerics).

The minimum-spanning tree (MST) was generated to evaluate the relationships between CCs and the source/region of isolation. The MST showed a star-like shape with several branches (Fig. 3A and Fig. S3). CC1 contained the most isolates, and these isolates originated from diverse ecological sources, as follows: human stool (263 isolates), human milk (one isolate), human blood (six isolates), human vagina (one isolate), and dog stool (one isolate). The isolates in CC1 originated from a large variety of regions (including countries from all five continents). The isolates clustered in CC2 to CC9 were all from human stool; those in CC2, CC4, and CC9 were from China, those in CC8 were from the U.S.; those in CC6 and CC7 were from Japan; those in CC5 were from Italy; and those in CC3 were from Japan and Ireland. The two isolates in CC10 were from human milk (Italy) and the human vagina (unknown country of origin). Host-specific evolution has been observed in some bacterial and fungal species (58, 59). Almost all human and animal-derived isolates (except strain CACC_517) did not cluster within the shared STs or CCs, indicating the unique genotype profile of the human-derived isolates and the highly diverse population of *B. longum*.

This is consistent with a previous study by Yanokura et al., who found that although bifidobacterial isolates from piglets formed a shared cluster, the 11 isolates of animal origin (including adult pigs, infants, and calves) did not show any split and together formed cluster A (13). Moreover, similar results were also observed with some *Lactobacillus* spp. (e.g., *Lc. lactis*, *Lb. sanfranciscensis*, *Lb. fermentum*, and *Lb. casei*) (24, 60) using MLST analysis. However, this is not consistent with the isolates of *Lb. plantarum* and *Lb. helveticus* (61, 62).

The number of STs identified in the present study is far greater than that observed in a previous study (13). One possible reason is that we included a vast number of *B. longum* isolates from diverse global sources. The high number of STs indicates the high heterogeneity and genetic diversity in *B. longum* isolates. Moreover, commercially available *B. longum* strains such as W11, KACC 91563, CECT 7347, 51A, BORI, W11, and 35624 were also included in this study (63–65). Except for strain 35625, which was assigned to ST60 containing multiple strains, other strains were all found to be singleton STs. By characterizing the genotype of the isolates, more isolates with potential applications in the food industry can be identified.

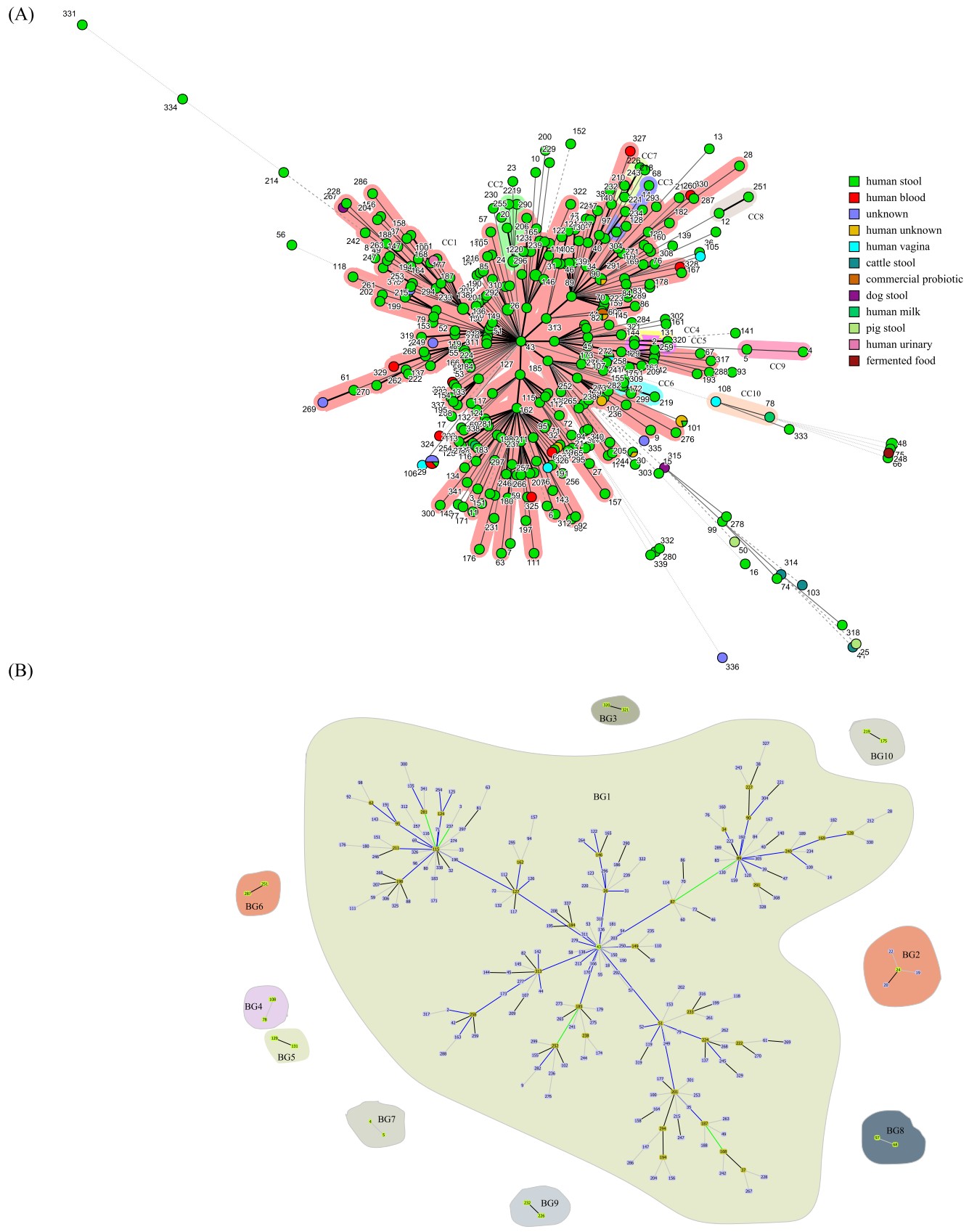

**FIG 3** Minimum spanning tree (MST) analysis of 437 *B. longum* isolates based on the allelic profiles of six MLST loci according to isolation source (A), and Clustering of BURST groups (BGs) based on the allelic profiles of the six MLST loci using the PHYLOViZ software (B). For the MST analysis using

Using traditional culture method, 30 *B. longum* isolates were recovered from the guts of the elderly. MLST genotyping based on six loci showed that the 30 *B. longum* strains could be typed into 24 STs. However, there were no shared STs between intestinal isolates from Enshi and Xiangyang. Using Prims's algorithm, we found that two isolates from Enshi city and four isolates from Xiangyang city could be clustered into CC1. Moreover, eight and two isolates from Xiangyang city were clustered into CC2, and CC9, respectively, and the other isolates were all singleton STs. CC2 and CC9 only contained the isolates from Xiangyang City, indicating that they may be two geographically specific CCs. In addition, genetic diversity appeared to be lower for isolates of Enshi origin, and there were no shared STs and CCs among the isolates from the two regions, likely owing to dietary differences. However, because few isolates from Enshi city were included in the MLST analysis, differentiating between the genetic diversity of *B. longum* isolates from the elderly in Enshi and Xiangyang city is challenging.

**Recombination analysis.** We performed split decomposition analysis, and results revealed that all six loci showed network-like structures with parallelograms (Fig. S4), indicative of recombination during evolutionary history. However, recombination was confirmed only at two loci—*clpC* and *rplB*—by the phi-test ($P < 0.05$). The other four loci—*fusA*, *ileS*, *purF*, and *rpoB*—had undergone little intergenic recombination during evolution ($P > 0.05$ on the phi test; see Table 1). In addition, the concatenated sequences of the six loci of the 341 STs showed parallelogram structures, suggesting the presence of recombination events ($P < 0.001$). For intestinal isolates from the elderly in Xiangyang city, recombination was confirmed ($P < 0.05$; Fig. S4). However, due to a limited number of isolates, no results were obtained for isolates of Enshi origin. Moreover, recombination was further determined based on the six MLST loci. The r/m values for all intestinal isolates of *B. longum* from the elderly were determined to be 6.53 and 0.90, respectively. Specifically, a 213-bp recombinant hot spot was detected only in STs of Xiangyang origin (ST19, 20, 22, 23, and 24), but no other recombination fragments were detected among the isolates from the elderly (Fig. S5).

MLST results showed a highly diverse population of *B. longum* isolates. In our study, recombination caused six times more polymorphisms than mutations. It seemed to be a key driver of evolution in *B. longum* from different regions worldwide and different isolation sources, indicating that *B. longum* is not a clonal species. The adaptive process could shift the selective pressure exerted on any adapting lineage, resulting in a higher or lower rate of recombination and/or mutation (66). The relative contribution of recombination and mutation to the genotype divergence of *B. longum* isolates from the elderly was found to be almost equally.

Notably, a recombinant fragment was detected in CC9, which was geographically specific. However, no shared recombination fragments were detected among the isolates from both Enshi and Xiangyang city. Recombination and mutation are major drivers of evolution in bacterial populations, and they contribute to the evolution of most bacteria (67, 68). Recombination events seemed to contribute to the formation of the specific genotype profile of isolates of Xiangyang origin. The presence of mobile genetic elements in the genome may contribute to such recombination events in *B. longum* (69, 70).

**Phylogenetic analysis.** The split decomposition tree for the concatenated sequences showed that the 341 STs could be divided into three lineages. These three lineages corresponded well to the three subspecies of *B. longum* (12). As shown by Albert et al., the subcluster B1-2 (containing the type strain *B. longum* subsp. suillum Su-851) did not correspond to any cluster in the MST or lineage in the split tree (Fig. 3 and 4).

**FIG 3** Legend (Continued)
BioNumerics, each circle is marked by the sequence type (ST) number, and the size of the circle is proportional to the number of isolates. STs belonging to the same clonal complex are indicated through surrounding shading. The color of the circle indicates the source of isolation. BGs were identified at the DLV threshold. In Fig. 3B, the number in each circle corresponds to the sequence type (ST). The circles representing the minor BGs (containing only two STs) are all colored yellow, while those representing the major BGs (containing three or more STs) are colored light purple (nonfounders), goldenrod (founders), and yellow (primary founders), respectively. The singleton STs were not shown.

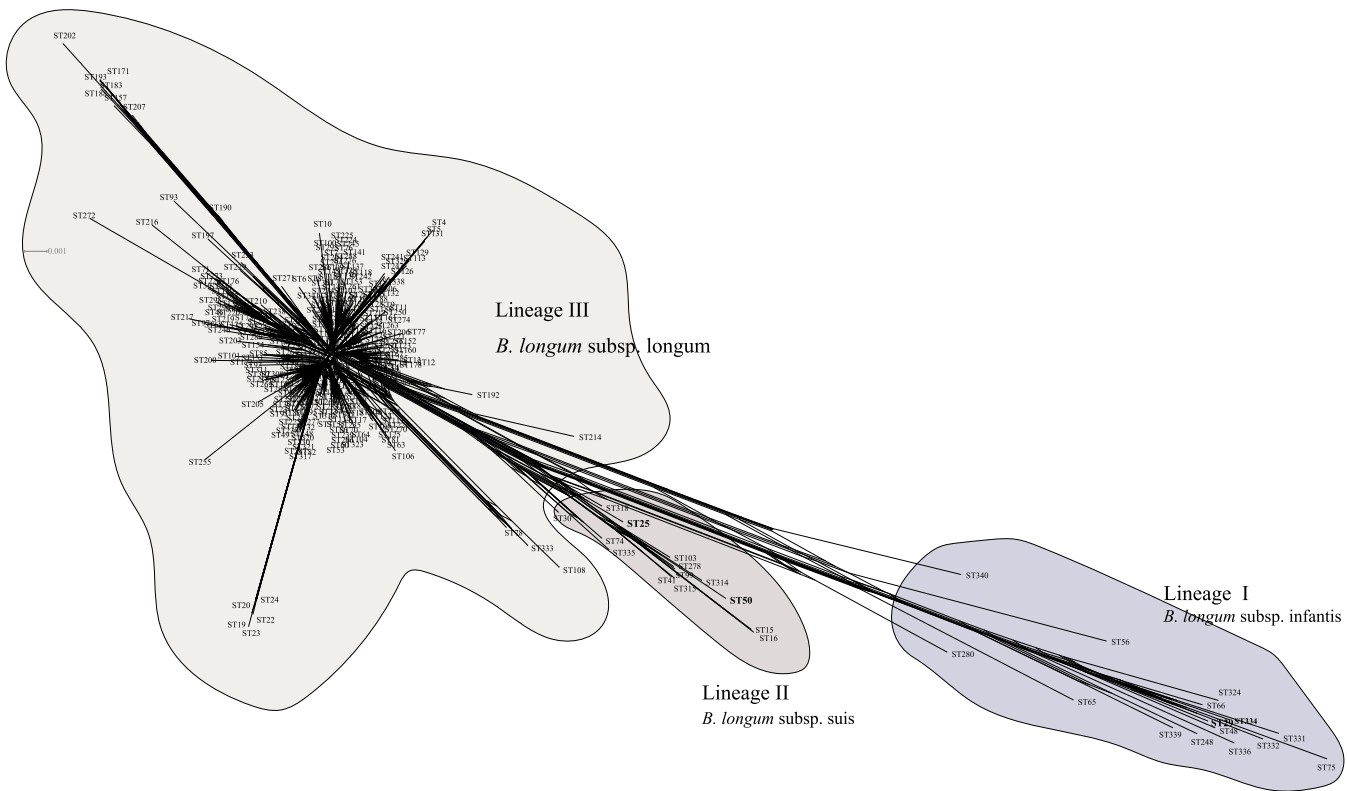

**FIG 4** Split decomposition analysis based on concatenated sequences of the six MLST loci. SplitsTree v4 was used for this analysis.

Meanwhile, ST25 (consisting of type strain Su-851) was not separated from cluster 3, indicating that Su-851 maybe a member of *B. longum* subsp. suis (14). We inferred that this discrepancy may be due to the limited number of isolates of animal origin (13).

Neighbor-joining (NJ) and maximum-likelihood (ML) trees were built from the concatenated sequences of the 341 STs to understand phylogeny (Fig. 5). Both trees showed that the 437 isolates could be divided into three lineages (corresponding to the three subspecies *B. longum* subsp. infantis, *B. longum* subsp. suis, and *B. longum* subsp. longum). In addition, ST50 (containing type strain DSM20211) and ST25 (type strain Su-851) formed a shared branch, indicating that Su-851 (type strain of *B. longum* subsp. suillum) should not be considered a novel subspecies. The topology of the species tree (Fig. S6) based on orthologous genes from the 407 genomes of *B. longum* was consistent with the phylogeny derived from the concatenated sequences (Fig. 5). Furthermore, the split network of the 341 STs also revealed three distinct subgroups corresponding to lineages I, II, and III, and ST50 and ST25 clustered together in lineage II (Fig. 4). This is consistent with a previous genome sequence-based report by Albert et al. (14).

From the phylogeny analysis based on the concatenated sequences of the six loci, all the three isolates from the elderly in Enshi city were classified as *B. longum* subsp. longum, and most isolates (25 out of 27 isolates) from Xiangyang city were classified as *B. longum* subsp. longum. However, only two isolates (HBUAS55030 and HBUAS55032) from the elderly in Xiangyang city were identified to be *B. longum* subsp. suis. Hence, the difference in the daily diet may also have affected the distribution of intestinal *B. longum* at the subspecies level. Furthermore, this is the first report of *B. longum* subsp. suis from the human gut (12, 13). It is noteworthy that no isolates from the elderly in this study were classified as *B. longum* subsp. infantis.

Furthermore, in the concatenated sequence-based phylogeny analysis, most *B. longum* subsp. infantis isolates were found to have originated from human sources (human gut and blood). Using MLST analysis, Makino et al. showed that *B. longum* subsp. longum isolates from the feces of a mother and her infant were monophyletic (18). However, isolates were only

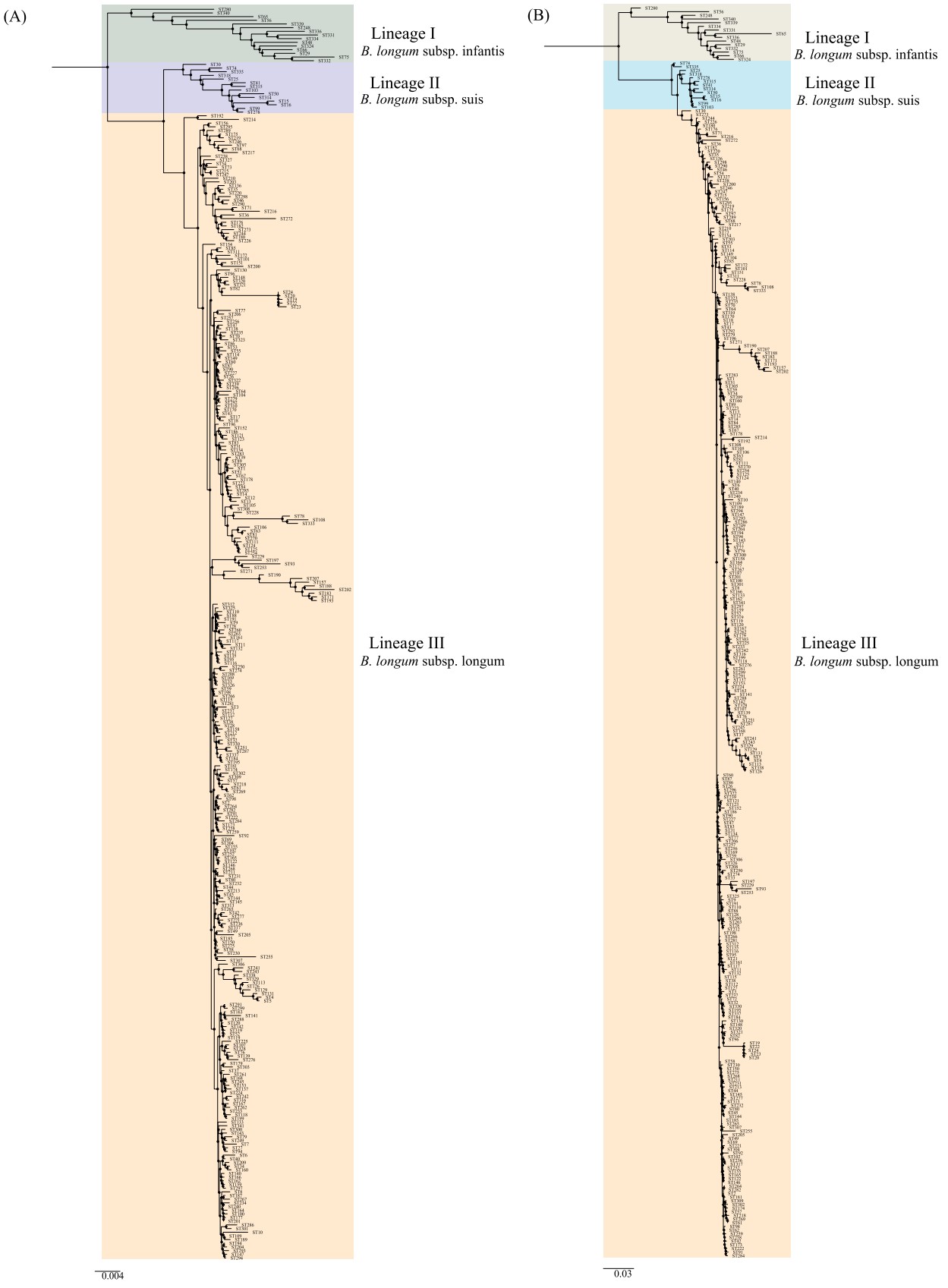

**FIG 5** Phylogenetic trees based on the six MLST loci in *B. longum* constructed using the neighbor-joining (NJ, A) and maximum likelihood (ML, B) methods. Filled circles indicate the tree nodes. For the NJ and ML phylogenetic trees, MEGA X software was used with the Kimura 2 parameter and GTRGAMMAIX model, respectively. The number of bootstrap replicates was set at 1,000. The two trees were visualized using Figtree V1.4.3 (http://tree.bio.ed.ac.uk/software/figtree/).

obtained from the mother and the infant. In our study, most of isolates that were classified as *B. longum* subsp. longum also came from humans, and the isolation sources were diverse and included human blood, milk, stools, urine, and the vagina. A total of nine isolates were classified as *B. longum* subsp. suis; these nine isolates formed a separate branch and were found to have both animal (including dog, pig, and cattle stools) and human origin, indicating that *B. longum* subsp. suis. is a non-host-specific subspecies, similar to *B. animalis* subsp. lactis (71). Previous studies recommend that multilocus sequence analysis be used as an alternative to 16S rRNA gene analysis, and that it be used for the species-level identification of *Bifidobacterium* (13). In this study, we confirmed that concatenated sequence-based phylogeny analysis with MLST can be used to infer taxonomic positions at the subspecies level in *B. longum*.

**Conclusions.** The diversity of the gut microbiome in the elderly from Xiangyang city and its neighboring region Enshi city was investigated in this study. High-throughput sequencing revealed a distinct difference in bacterial diversity between the two groups. However, both groups showed similar relative abundance of bifidobacteria. Using traditional culture method, we showed that the isolation rate of *B. longum* is lower in the guts of elderly individuals from Enshi city than in those from Xiangyang city. We speculated that diet were responsible for differences in the abundance of *B. longum* between the two cities. A total of 30 *B. longum* isolates were used for MLST analysis along with other 407 previously identified *B. longum* genomes. MLST analysis suggested a high genetic diversity in *B. longum* isolates. Twenty-four STs were identified among the 30 *B. longum* isolates from the elderly. MLST analysis revealed two putative geographically specific CCs of Xiangyang origin. The relative contribution of recombination and mutation to the genotype divergence of *B. longum* isolates from the elderly was almost equal. A unique recombinational fragment was detected in one of the putative geographically specific CCs, indicative of the possible contribution of recombination to the genotypic profile. Phylogenetic analysis showed that the 437 *B. longum* isolates could be divided into three *B. longum* subspecies. Most of the isolates from the elderly were identified to be *B. longum* subsp. longum, although two isolates of Xiangyang origin were classified as *B. longum* subsp. suis. The six MLST loci were effective in the classification of *B. longum* isolates. Our results revealed that differences in the daily diet influence intestinal bacterial diversity and the abundance of *B. longum*. Our findings could contribute to future studies on the discrimination of *B. longum*.

## MATERIALS AND METHODS

**Sample collection.** A total of 42 participants were included in this study; 20 participants (10 women, 10 men) were from Xiangyang city (32.00N°, 112.18°E) and 22 participants (11 women, 11 men) were from Enshi city (30.28°N, 109.48°E; Fig. S1). These participants were aged 64 to 90 years and had no major medical issues or history of antibiotic use within the previous 3 months. This study was approved by the ethic committee of the Hubei University of Arts and Science (No. 2020-009), and we obtained the consent from the participants.

Stool samples were collected by the participants using a plastic stool-collecting container. A portion of the fresh stool sample was placed into a 50-mL tube with a gas exchange lid and then quickly placed into an anaerobic jar containing an AnaeroGen Gas pack. Another portion was immediately frozen and then quickly transferred to a food microbiology laboratory where it was stored at $-70$°C.

**PCR amplification and MiSeq amplicon sequencing.** Total DNA was extracted from 5 g of stool samples using an E.Z.N.A. Stool DNA Kit (Omega Bio-tek, USA), according to the manufacturer's protocol. Once DNA was extracted, the quantity and purity of the extracts were assessed using a Nanodrop 2000 machine (Thermo, USA) and electrophoresis (0.8% agarose gel). The samples were then stored at $-20$°C until further use. The 16S rRNA V3-V4 region was PCR amplified using the barcoded adaptor-containing forward 338F (5′- barcode ACT CCT ACG GGA GGC AGC A-3′) and reverse 806R (5′- GGA CTA CHV GGG TWT CTA AT -3′) primers, as previously described (72). To explore the bacterial population, MiSeq sequencing targeting the 16S rRNA V3-V4 region was performed. The resulting amplicons were normalized and sequenced via an Illumina MiSeq platform (San Diego, CA, USA) to generate paired-end raw reads.

**Bioinformatic analysis.** Paired-end sequence reads were analyzed using the Quantitative Insights Into Microbial Ecology (QIIME) platform v1.91 (73). First, the low-quality reads, primers, and barcode sequences were filtered out. The filtered paired-end reads were merged based on overlapping sequences using FLASH (74). The merged assembly sequences were used to generate an operational taxonomic unit (OTU) using uCLUST based on an identity threshold of 97% (75). Taxonomy assignment was performed by blasting representative OTU sequences using a combination of three databases: the latest Greengenes database (76), Ribosomal Database Project (77), and SILVA (78).

Statistical analyses were conducted using nonparametric tests (Wilcoxon test and Kruskal-Wallis test) in RStudio v3.6.1. $\alpha$-diversity indices—including Chao1, observed species, Shannon, and Simpson indices—and sequencing depth were calculated using QIIME. The $\beta$-diversity of stool samples from the elderly living in

different regions was analyzed using R package (vegan v2.5-6) (https://cran.r-project.org/web/packages/vegan/index.html). $\alpha$- and $\beta$-diversity were visualized using R package ggplot and ggpubr, respectively. Difference in overall bacterial community composition between the two regions were assessed using PD functions in R package picante (79). The linear discriminant analysis (LDA) effect size (LEfSe) method (http://huttenhower.sph.harvard.edu/lefse/) (80) was used to determine taxa biomarkers (default parameters, i.e., stricter model with no correction), and the LDA threshold score was set at 3.0.

**Isolation and identification of bifidobacterial strains.** Ten grams of each stool sample from the anaerobic jar were blended with 90 mL sterile physiological saline and vortexed thoroughly to separate the bacteria from organic particles. The mixture was serially diluted 10-fold, and 100 $\mu$L of the dilutions ($10^{-2}$, $10^{-3}$, $10^{-4}$, $10^{-5}$, and $10^{-6}$) was plated onto mMRS agar (MRS containing 5% sheep blood, 0.05% [v/w] L-cysteine hydrochloride, and mupirocin [50 mg/L]) (81). The plates were placed into a DG250 anaerobic system (Don Whitely Scientific; United Kingdom) flushed with a gas mixture (85% $N_2$, 10% $CO_2$, and 10% $H_2$) and incubated at 37°C for 72 h (82). Almost all colonies were picked and purified by restreaking onto mMRS agar at least thrice. The purified isolates were observed by optical microscopy to examine morphology and Gram stain positivity. Additionally, they were tested for catalase activity. All Gram-positive and catalase-negative isolates were further evaluated for fructose-6-phosphate phosphoketolase (F6PPK) activity, as described by Chen et al. (83). The potential bifidobacterial isolates that showed F6PPK positivity were preserved in mMRS broth containing 30% glycerol at $-70$°C.

These possible bifidobacterial isolates were further evaluated using 16S rRNA sequencing. The 16S rRNA gene was amplified and sequenced as previously described (82). The PCR product amplified using 27F and 1495R primers was purified using an AxyPrep PCR Clean-up kit (Axygen, Hangzhou, China), ligated into the T-vector pMD18-T (TaKaRa, Dalian; China), and transformed into competent DH5$\alpha$ *E. coli* cells. The plasmid was extracted from positive clones and sequenced on a 3730 ABI sequencer (service provided by WuHan Tianyihuiyuan Biotechnology Co., Ltd., Wuhan, China). A nearly complete 16S rRNA sequence for each possible bifidobacterial isolate was obtained through the assembly of the forward and reverse sequences and removal of amplification primer sequences. These sequences were then inputted into the NCBI blastn database. Subsequently, a NJ phylogenetic tree was created for these isolates using MEGA vX (84), and the tree was visualized using the R package ggtree (85).

**MLST analysis.** Following the protocol reported by Delétoile et al., six housekeeping loci—ATP-dependent Clp protease ATP-binding subunit (*clpC*); GTP-binding protein chain elongation factor (*fusA*); isoleucyl-tRNA synthetase (*ileS*); amidophosphoribosyltransferase (*purF*); 50S ribosomal subunit protein L2 (*rplB*); and the beta-subunit of RNA polymerase (*rpoB*)—were selected for the MLST analysis of *B. longum* isolates. The PCR primers and thermal cycling conditions for amplifying these six genes were modified based on previous literature (17). The PCR mixture (50 $\mu$L) contained 0.5 $\mu$L PrimeSTAR HS DNA polymerase (5 U/$\mu$L, TaKaRa, Tokyo), 10 $\mu$L 5 $\times$ PCR buffer (Mg$^{2+}$ Plus), 4.0 $\mu$L dNTPs (2.5 mM each), 1.0 $\mu$L forward primer (10 $\mu$M), 1.0 $\mu$L reverse primer (10 $\mu$M), 1 $\mu$L genomic DNA (10 to 50 ng/$\mu$L), and dH$_2$O to make up the reaction volume to 50 $\mu$L. PCR products were visualized by 1.2% agarose gel electrophoresis to determine the presence and size of amplified products. Then, the PCR products were purified using a DNA purification kit (Axygen, Hangzhou, China) and sequenced, as was done for the PCR analysis of the 16S rRNA gene.

Sequences for the six loci in the 30 strains were assembled using BioNumerics v7.6 and added to the NCBI database for examination. In order to compare MLST sequence data of the strains examined in this study, the genome of *B. longum* was downloaded from the NCBI database and the sequences for the six housekeeping genes were extracted using TBtools v1.086 (86). The ANI was calculated using pyANI v0.3.0 (https://github.com/widdowquinn/pyani) to eliminate the strains that were not *B. longum*. The GC content and length of the genome were determined using seqkit v0.15.0 (87).

Sequences for each of the six housekeeping loci were loaded onto an MLST database established in BioNumerics v7.6, and these sequences were aligned using ClustalW and trimmed to a uniform length manually. Allele numbers and STs were assigned randomly using default parameters in BioNumerics v7.6. A MST was constructed using Prims's algorithm in BioNumerics v7.6 according to the source or region of isolation. The clustering of *B. longum* STs was performed using allelic profiles as input and PHYLOViZ v2.0 software through the goeBURST algorithm (57). The mean GC content, nucleotide diversity, and number of polymorphic sites and single nucleotide polymorphisms in the six partial housekeeping gene fragments were calculated using DnaSP 5.0 (88). The dN/dS ratio was calculated using the CodeML function in the paml v4.7j package (89). The phi test for recombination, generation of the split decomposition tree for each individual locus, and the concatenation of sequences were performed using SplitsTree 4.0 (90). NJ and ML trees were constructed based on the concatenated sequences using MEGA vX and RAxML v8 (84, 91), respectively. ClonalFrameML v1.12 (92) and Gubbins v2.4.1 (93) were used to evaluate the recombination at the six MLST loci. The kappa value was calculated using the paml v4.7j package and used as input for ClonalFrameML, while the results from Gubbins were visualized by using Phandango (94).

**Data availability.** The high-throughput sequence data are available at MG-RAST (http://www.mg-rast.org/) under accession number of mgp98194. The 16S rRNA gene sequences for the bifidobacteria are available at GenBank database under the accession numbers of MH685117-MH685166, and MH681640-MH681651. The accession for the six loci of *B. longum* isolates at GenBank database are MT810124 -MT810303. The supplementary material for this article can be found online at: https://doi.org/10.6084/m9.figshare.14397692.v1.

## SUPPLEMENTAL MATERIAL

Supplemental material is available online only.
**SUPPLEMENTAL FILE 1**, XLSX file, 0.01 MB.
**SUPPLEMENTAL FILE 2**, XLSX file, 0.01 MB.

SUPPLEMENTAL FILE 3, XLSX file, 0.04 MB.
SUPPLEMENTAL FILE 4, PDF file, 1.1 MB.

## ACKNOWLEDGMENTS

Z.Z. and Z.G. designed the experiment; Z.Z., Z.G., Y.W., and F.X. prepared the stool samples; Q.H., Z.Z., and Z.G. analyzed the data; and Z.Z., Y.W., and F.X. performed the experiments; Z.Z. wrote the manuscript; Z.G. reviewed the manuscript. All authors read and approved the manuscript.

This work was supported by the National Natural Science Foundation of China (31501455), Medical and Health Science and Technology Development Plan of Xiangyang (2020ZD10), and Hubei University of Arts and Science Foundation for Cultivation Fund for Teachers' Scientific Research Ability: "Technological Innovation Team" (2020kypytd009).

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
