## [Reviewer comments · Microbiology Spectrum]

Microbiology Spectrum

Evaluation of bacterial diversity and evolutionary dynamics of gut *Bifidobacterium longum* isolates obtained from older individuals in Hubei Province, China

Zhendong Zhang, Qiangchuan Hou, Yurong Wang, Fanshu Xiang, and Zhuang guo

Corresponding Author(s): Zhuang guo, Hubei University of Arts and Science

Review Timeline:

Submission Date:	September 1, 2021
Editorial Decision:	November 17, 2021
Revision Received:	December 1, 2021
Editorial Decision:	December 10, 2021
Revision Received:	December 13, 2021
Accepted:	December 23, 2021

Editor: Steven Frese

Reviewer(s): Disclosure of reviewer identity is with reference to reviewer comments included in decision letter(s). The following individuals involved in review of your submission have agreed to reveal their identity: Rustem Abuzarovich Ilyasov (Reviewer #2)

Transaction Report:

DOI: <https://doi.org/10.1128/Spectrum.01442-21>

November 17, 2021

Dr. Zhuang guo
Hubei University of Arts and Science
No. 296 Longzhong Road Xiangyang, Hubei, China
Xiangyang
China

Re: Spectrum01442-21 (**Evaluation of bacterial diversity and evolutionary dynamics of gut *Bifidobacterium longum* isolates obtained from older individuals in Hubei Province, China**)

Dear Dr. Zhuang guo:

Please see the attached review comments as well.

Link Not Available

Sincerely,

Steven Frese

Journals Department
Editor Comments:

Line 58-60: Bifidobacteria are not the predominant microbes in the human gut, and rarely account for more than a few percent of the gut microbiome in adults.

Line 103: There is no IRB approval, and the manuscript cannot be published without this. Please note IRB approval prior to study start and note the appropriate documentation or identifier for this approval.

Line 144-149: Lefse is noted as a means to determine differences between samples, but no model is provided. Please note which parameters were used to correct for differences in the linear discriminant analysis.

Line 295-298: The statement that there are more diverse fermented and pickled vegetables as a result of altitude is not logical. Please add clarifying language.

Line 303-316: Enterotyping is noted as a discriminatory measure of sample microbiomes, but no methods are provided.

Line 322-323: The sentence does not make sense as written. There is no conclusion about barrier function that can be drawn

from the preceding statements.

Line 326-329: Please cite the primary literature supporting these statements.

Line 348-349: Please cite the statement that *B. longum* is widely reported to play a role in host health and disease susceptibility.

Line 350-351: The abundance of *B. longum* cannot be stable over time and vary with age, the statement contradicts itself.

Line 400: Please also include the more common strain designation, ATCC 15679.

Line 455: "there may be two geographically specific CCs" - The definitive 'are' is not completely supported, please soften the language to 'may'

Line 488-490: The evidence provided may only apply to *B. longum*, please note this.

Line 509- 520: If your phylogenetic tree predicts subspecies, please note them throughout the figures and text as predicted subspecies.

Line 521: No evidence is provided that confirms that the gut of the elderly is a reservoir of *B. longum*. The organism may be widely found, but the present study only measured the elderly.

Line 533-538: Most of the reported 'blood' derived *B. longum* isolates may have originated from probiotic administration. Please confirm that the genomes used either originated from probiotic related sepsis, or that they were indeed distinct.

Line 538-542: No transmission is demonstrated for these strains in the present study, please remove the statement.

Line 554-555: The relationship between diet and city-specific differences is speculative as no dietary information is provided. Please note the comment as being speculative.

Line 559: You did not observe a shared genotype, but that does not mean it is not there. Please soften the language to address this.

Line 563: As above, a lack of observation does not generate a conclusion that it is indeed geographically specific. Please note that this is speculative.

Reviewer comments:

Reviewer #2 (Comments for the Author):

Manuscript Spectrum01442-21 - Evaluation of bacterial diversity and evolutionary dynamics of gut *Bifidobacterium longum* isolates obtained from older individuals in Hubei Province, China

The authors characterized intestinal bacterial diversity and evolution of *Bifidobacterium longum* of elderly origin and found differences in bacterial populations in the guts of the elderly between Xiangyang and the neighboring Enshi city. The authors showed the *B. longum* predominates throughout the lifespan, from birth to old age, and could alter the intestinal microbial population and immune function in the elderly. This work is interesting and important for further studies on genotyping and discrimination of *B. longum* in the human gut.

The statistical methods were used properly. The experiment was provided correctly.

All the authors' conclusions are supported by their data.

The English of the text is well, but should be checked by a professional translator.

The uniqueness of the text is more than 90% by AntiPlagiarism.NET.

The text contains some misspellings and typos:

In the Abstract the sentence "A total of 62 bifidobacterial strains were isolated, of which 30 isolates were found to be *B. longum*, and fewer isolates were recovered from the elderly in Enshi city. " should be corrected.

Lint 97 - "the aim of this study was" should be "this study aimed"

Line 105 - Comment for authors: The number of participants is 42 is enough for statistics in this experiment, but can be increased in further studies.

Line 291 - for the sentence "The intestinal microbiome is an important determinant of human health, and its diversity can be influenced by diet, lifestyle, diseases, and medication use (49, 50)" add additional citation (Ilyasov et al., 2012) and add to the references Ilyasov, R.A.; Gaifullina, L.R.; Saltykova, E.S.; Poskryakov, A.V.; Nikolenko, A.G. Review of the expression of antimicrobial peptide defensin in honey bees *Apis mellifera* L. *Journal of Apicultural Science* 2012, 56, 115-124. doi: 10.2478/v10289-012-0013-y

Line 355 - "high-though sequencing" should be "high-throughput sequencing"

Line 483 - "is a not a clonal" should be "is not a clonal"

Line 536 - "most isolates that were" should be "most of isolates were"

Line 553 - to the sentence "lower isolation rate of *B. longum* isolation in the gut of the elderly from Enshi city." add "compared to Xiangyang city"

Line 568 - "The six MLST loci is effective" should be "The six MLST loci are effective"

Line 570 - "difference of daily diet " should be "difference in daily diet "

Line 589 - "approved by approved by" should be "approved by"

Please correct the manuscript according to the above comments.

I attached the pdf with highlighted comments.

Editor Comments:

Please carefully edit the manuscript for typos (e.g., Line 42: "Induction").

Line 58-60: Bifidobacteria are not the predominant microbes in the human gut, and rarely account for more than a few percent of the gut microbiome in adults.

Line 103: There is no IRB approval, and the manuscript cannot be published without this. Please note IRB approval prior to study start and note the appropriate documentation or identifier for this approval.

Line 144-149: Lefse is noted as a means to determine differences between samples, but no model is provided. Please note which parameters were used to correct for differences in the linear discriminant analysis.

Line 295-298: The statement that there are more diverse fermented and pickled vegetables as a result of altitude is not logical. Please add clarifying language.

Line 303-316: Enterotyping is noted as a discriminatory measure of sample microbiomes, but no methods are provided.

Line 322-323: The sentence does not make sense as written. There is no conclusion about barrier function that can be drawn from the preceding statements.

Line 326-329: Please cite the primary literature supporting these statements.

Line 348-349: Please cite the statement that *B. longum* is widely reported to play a role in host health and disease susceptibility.

Line 350-351: The abundance of *B. longum* cannot be stable over time and vary with age, the statement contradicts itself.

Line 400: Please also include the more common strain designation, ATCC 15679.

Line 455: "there may be two geographically specific CCs" - The definitive 'are' is not completely supported, please soften the language to 'may'

Line 488-490: The evidence provided may only apply to *B. longum*, please note this.

Line 509- 520: If your phylogenetic tree predicts subspecies, please note them throughout the figures and text as predicted subspecies.

Line 521: No evidence is provided that confirms that the gut of the elderly is a reservoir of *B. longum*. The organism may be widely found, but the present study only measured the elderly.

Line 533-538: Most of the reported 'blood' derived *B. longum* isolates may have originated from probiotic administration. Please confirm that the genomes used either originated from probiotic related sepsis, or that they were indeed distinct.

Line 538-542: No transmission is demonstrated for these strains in the present study, please remove the statement.

Line 554-555: The relationship between diet and city-specific differences is speculative as no dietary information is provided. Please note the comment as being speculative.

Line 559: You did not observe a shared genotype, but that does not mean it is not there. Please soften the language to address this.

Line 563: As above, a lack of observation does not generate a conclusion that it is indeed geographically specific. Please note that this is speculative.

Staff Comments:

Preparing Revision Guidelines

Please return the manuscript within 60 days; if you cannot complete the modification within this time period, please contact me. If you do not wish to modify the manuscript and prefer to submit it to another journal, please notify me of your decision immediately so that the manuscript may be formally withdrawn from consideration by Microbiology Spectrum.

By profiling *Bifidobacterium* isolates in two elderly populations, this paper begins to identify genomic diversity amongst *B. longum* subsp. *longum* strains in the Hubei Province, China. This paper provides microbial community profiling for a population not frequently explored. While phylogenetic diversity analysis revealed differences between the two cities (line 280) the implications of such differences requires further investigation.

Diet is frequently referred to as the putative reason for regional microbial community differences (line 554 and line 570) but no data is provided to support such claims. The paragraph describing some of the regional diets (line 295-302) was helpful in outlining generalizable differences between the groups. However, metadata on diet, birthplace, health outcomes, etc. is necessary to establish such claims about diet considering how individualistic and resource-driven microbial composition can be.

A few areas that could begin to address this concern would be genomic carbohydrate utilization analysis to provide further evidence of dietary-driven enrichments. Whole genome sequencing of selected strains would be advantageous to contrast the functional capabilities between strains. A comment is made about selenium dietary differences (line 299), further investigation into selenium-associated genes or selenium *in vitro* growth would provide evidence to this claim. Additionally, a paragraph outlining the limitations of this research may be required.

Considering previous associations between *B. longum* subsp. *longum* and genes related to plant-glycan utilization, genomic analysis or *in vitro* screening of plant carbohydrate degradation is appropriate especially as diet is being hypothesized as the differentiation factor (line 366).

The LEfSe analysis highlights a few taxa as being differential between the two cities. Was any qPCR conducted to establish whether these taxa were significantly different by absolute abundance? Such data would strengthen claims regarding distinct microbial communities. Based on the *Escherichia/Shigella* differential taxa abundance and comments regarding proinflammatory states (line 316) qPCR of inflammation/pathogen-associated microbial clades may bolster findings.

This research raises further questions related to the transmission and extinction of *Bifidobacterium* taxa within regionally isolated communities. A profile of elderly populations is useful as a comparison and a larger investigation of these cities may provide further insight into how *B. longum* species are transmitted across generations.

Minutia:

Line 117: Did the DNA extraction protocol include a mechanical lysis step? This has been found to impact *Bifidobacterium* DNA yield. Walker, A.W., Martin, J.C., Scott, P. *et al.* 16S rRNA gene-based profiling of the human infant gut microbiota is strongly influenced by sample processing and PCR primer choice. *Microbiome* **3**, 26 (2015). <https://doi.org/10.1186/s40168-015-0087-4>

Line 69: Ensure all subspecies names are italicized.

Line 131: Were the samples rarified during QIIME processing? Considering the alpha diversity comparisons by region, I expected a note about rarefaction to be included in the methods section.

Line 241: A comment is made regarding microbial diversity being a measure of health. Were any health outcomes actually measured between these populations that would suggest the microbial diversity was a marker for a “healthier enterotype?”

Line 280: What statistical test was conducted? Was a PERMANOVA of the Bray Curtis analysis conducted for the data visualized in 1d?

Lines 342-345: Numbers may be better understood in a table format.

Line 358: clarify which groups of 20 and 22 you are referencing.

Annotation for Figure 5 Legend could be clearer as to what each figure is (e.g. “constructed using (A) the neighbor-joining and (B) maximum likelihood methods”).

Line 497: where these mobile genetic elements also present in the Enshi isolates? Based on wording, uncertain as to whether Xingyang isolates were being identified as specifically having more mobile elements.

Line 529: why is this noteworthy? An explanation may be helpful for the reader.

Line 49: Change to “introduction.”

Response to Reviewers

We would like to thank the reviewers for their thoughtful comments and suggestions for the manuscript. We hope that our responses adequately address the comments, and the final revisions are acceptable. Our point-by-point responses to these comments are as follows:

Editor Comments:

Line 58-60: Bifidobacteria are not the predominant microbes in the human gut, and rarely account for more than a few percent of the gut microbiome in adults.

Response: Thank you very much for your comments. These have been revised.

Line 103: There is no IRB approval, and the manuscript cannot be published without this. Please note IRB approval prior to study start and note the appropriate documentation or identifier for this approval.

Response: Thank you very much for your comments. An IRB approval material was provided and in this submission.

Line 144-149: Lefse is noted as a means to determine differences between samples, but no model is provided. Please note which parameters were used to correct for differences in the linear discriminant analysis.

Response: Thank you very much for your comments. In LEFSe analysis, there are three arguments, "0,1,2", for function "mult.test.correction". In our data analysis, we used the default parameter "0", namely "no correction (a more strict model with no correction)"

Line 295-298: The statement that there are more diverse fermented and pickled vegetables as a result of altitude is not logical. Please add clarifying language.

Response: Thank you very much for your comments. These have been revised in line 295- 300. Enshi city is located in a mountainous region with a higher altitude, and its population includes a higher number of Tujia and Miao people. In contrast, Xiangyang city has a lower altitude, and the residents are Han people. Therefore, compared to Xiangyang city, Enshi city has more diverse fermented and pickled vegetables with more pepper, including the typical traditional fermented food — Zha-chili, owing to differences in the lifestyles of its population.

Line 303-316: Enterotyping is noted as a discriminatory measure of sample microbiomes, but no methods are provided.

Response: Thank you very much for your comments. These have been revised.

Line 322-323: The sentence does not make sense as written. There is no conclusion about barrier function that can be drawn from the preceding statements.

Response: Thank you very much for your comments. These have been revised in line 323 - 326. The four genera were enriched in the elderly from Xiangyang. The four genera were enriched in the elderly from Xiangyang. These genera belong to the phylum Firmicutes, and specifically to clostridial clusters IV and XIVa, which are reported to

be butyrate-producers. Studies have shown that microbial butyrate increases gut barrier function. Hence, these gram-positive, strictly anaerobic, saccharolytic bacteria are speculated to be beneficial for maintaining gut barrier function.

Line 326-329: Please cite the primary literature supporting these statements.

Response: Thank you very much for your comments. These have been revised. and the corresponding references were cited in line 329 - 333, and added in Reference section.

Line 348-349: Please cite the statement that *B. longum* is widely reported to play a role in host health and disease susceptibility.

Response: Thank you very much for your comments. These have been revised in 352, and added in Reference section.

Line 350-351: The abundance of *B. longum* cannot be stable over time and vary with age, the statement contradicts itself.

Response: Thank you very much for your comments. This has been revised.

Line 400: Please also include the more common strain designation, ATCC 15679.

Response: Thank you very much for your comments. We searched the online NCBI database, however, genome and the housekeeping gene sequence of strain ATCC 15679 were not available by now.

Line 455: "there may be two geographically specific CCs" - The definitive 'are' is not completely supported, please soften the language to 'may'

Response: Thank you very much for your comments. This has been revised in line 455.

Line 488-490: The evidence provided may only apply to *B. longum*, please note this.

Response: Thank you very much for your comments. The statement was removed. Calculatation of r/m values for the same species was not found in the previous studies.

Line 509- 520: If your phylogenetic tree predicts subspecies, please note them throughout the figures and text as predicted subspecies.

Response: Thank you very much for your comments. These have been revised in Figures 4 and 5.

Line 521: No evidence is provided that confirms that the gut of the elderly is a reservoir of *B. longum*. The organism may be widely found, but the present study only measured the elderly.

Response: Thank you very much for your comments. This has been revised.

Line 533-538: Most of the reported 'blood' derived *B. longum* isolates may have originated from probiotic administration. Please confirm that the genomes used either originated from probiotic related sepsis, or that they were indeed distinct.

Response: Thank you very much for your comments. These have been revised. A total of 13 isolates from human blood were included in the MLST analysis. Results showed that five of the isolates were assigned to ST29, and other isolates only corresponded to a single isolate and were distinct from each other. We compared the five isolates in ST29, and all the five strains were highly similar and no unique region was found. The ANI (n average nucleotide identity) value for pairwise genome among the genomes of the five isolates are all higher than 99.99% (Figure d1). Therefore, we carried out an analysis of single-nucleotide polymorphisms (SNPs) using Parsnp v1.5.6 (<https://github.com/marbl/parsnp>). The SNP sites was extracted from VCF file using HarvestTools v1.3 (<https://github.com/marbl/harvest-tools>). The result of SNP analysis revealed a total of 172 SNP sites and several unique SNP sites for each of the five isolates (Figure d2). Although the similarity is high, this proves the heterogeneity of these isolates.

Figure d1. Average nucleotide identity (ANIb) heatmap. ANIb for all 13 genomes of human blood origin was calculated based on genome sequences using pyani 0.2.11.

Figure d2. Single-nucleotide polymorphisms (SNPs) sites of the five isolates of human blood origin in ST2

Line 538-542: No transmission is demonstrated for these strains in the present study, please remove the statement.

Response: Thank you very much for your comments. This has been revised in line 537-538.

Line 554-555: The relationship between diet and city-specific differences is speculative as no dietary information is provided. Please note the comment as being speculative.

Response: Thank you very much for your comments. This has been revised in 550-551.

Line 559: You did not observe a shared genotype, but that does not mean it is not there. Please soften the language to address this.

Response: Thank you very much for your comments. These have been revised.

Line 563: As above, a lack of observation does not generate a conclusion that it is indeed geographically specific. Please note that this is speculative.

Response: Thank you very much for your comments. This has been revised in line 558-560.

Reviewer comments:

Reviewer #2 (Comments for the Author):

Manuscript Spectrum01442-21 - Evaluation of bacterial diversity and evolutionary dynamics of gut *Bifidobacterium longum* isolates obtained from older individuals in Hubei Province, China. The authors characterized intestinal bacterial diversity and evolution of *Bifidobacterium longum* of elderly origin and found differences in bacterial populations in the guts of the elderly between Xiangyang and the neighboring Enshi city. The authors showed the *B. longum* predominates throughout the lifespan, from birth to old age, and could alter the intestinal microbial population and immune function in the elderly. This work is interesting and important for further studies on genotyping and discrimination of *B. longum* in the human gut. The statistical methods were used properly. The experiment was provided correctly. All the authors' conclusions are supported by their data.

The English of the text is well, but should be checked by a professional translator. The uniqueness of the text is more than 90% by AntiPlagiarism.

Response: Thank you very much for your comments. These have been revised.

NET. The text contains some misspellings and typos: In the Abstract the sentence "A total of 62 bifidobacterial strains were isolated, of which 30 isolates were found to be *B. longum*, and fewer isolates were recovered from the elderly in Enshi city. " should be corrected.

Response: Thank you very much for your comments. These have been revised.

Line 97 - "the aim of this study was" should be "this study aimed"

Response: Thank you very much for your comments. These have been revised.

Line 105 - Comment for authors: The number of participants is 42 is enough for statistics in this experiment, but can be increased in further studies.

Response: Thank you very much for your comments. In the future study, we would accommodate more participants.

Line 291 - for the sentence "The intestinal microbiome is an important determinant of human health, and its diversity can be influenced by diet, lifestyle, diseases, and medication use (49, 50)" add additional citation (Ilyasov et al., 2012) and add to the references Ilyasov, R.A.; Gaifullina, L.R.; Saltykova, E.S.; Poskryakov, A.V.; Nikolenko, A.G. Review of the expression of antimicrobial peptide defensin in honey bees *Apis mellifera* L. Journal of Apicultural Science 2012, 56, 115-124. doi: 10.2478/v10289-012-0013-y

Response: Thank you very much for your comments. These have been revised.

Line 355 - "high-though sequencing" should be "high-throughput sequencing"

Response: Thank you very much for your comments. These have been revised.

Line 483 - "is a not a clonal" should be "is not a clonal"

Response: Thank you very much for your comments. These have been revised.

Line 536 - "most isolates that were" should be "most of isolates were"

Response: Thank you very much for your comments. These have been revised.

Line 553 - to the sentence "lower isolation rate of *B. longum* isolation in the gut of the elderly from Enshi city." add "compared to Xiangyang city"

Response: Thank you very much for your comments. These have been revised.

Line 568 - "The six MLST loci is effective" should be "The six MLST loci are effective"

Response: Thank you very much for your comments. These have been revised.

Line 570 - "difference of daily diet " should be "difference in daily diet "

Response: Thank you very much for your comments. These have been revised.

Line 589 - "approved by approved by" should be "approved by"

Response: Thank you very much for your comments. These have been revised.

Please correct the manuscript according to the above comments.

I attached the pdf with highlighted comments.

Editor Comments:

Please carefully edit the manuscript for typos (e.g., Line 42: "Induction").

Response: Thank you very much for your comments. These have been revised.

Reference

1. Yan H, Ajuwon KM. 2017. Butyrate modifies intestinal barrier function in IPEC-J2 cells through a selective upregulation of tight junction proteins and activation of the Akt signaling pathway. *PLoS One* 12:e0179586.
2. Zhang L, Liu C, Jiang Q, Yin Y. 2021. Butyrate in energy metabolism: there is still more to learn. *Trends in Endocrinology & Metabolism* 32:159-169.

December 10, 2021

Dr. Zhuang guo
Hubei University of Arts and Science
No. 296 Longzhong Road Xiangyang, Hubei, China
Xiangyang
China

Re: Spectrum01442-21R1 (**Evaluation of bacterial diversity and evolutionary dynamics of gut *Bifidobacterium longum* isolates obtained from older individuals in Hubei Province, China**)

Dear Dr. Zhuang guo:

Thank you for submitting your manuscript to Microbiology Spectrum. As you will see your paper is very close to acceptance. Please modify the manuscript along the lines I have recommended. As these revisions are quite minor, I expect that you should be able to turn in the revised paper in less than 30 days, if not sooner. If your manuscript was reviewed, you will find the reviewers' comments below.

When submitting the revised version of your paper, please provide (1) point-by-point responses to the issues I raised in your cover letter, and (2) a PDF file that indicates the changes from the original submission (by highlighting or underlining the changes) as file type "Marked Up Manuscript - For Review Only". Please use this link to submit your revised manuscript. Detailed instructions on submitting your revised paper are below.

Link Not Available

Sincerely,

Steven Frese

Reviewer comments:

The IRB approval must be noted in the Methods section of the manuscript with the approval number and/or approval date.

Preparing Revision Guidelines

- point-by-point responses to the issues I raised in your cover letter
- Upload a compare copy of the manuscript (without figures) as a "Marked-Up Manuscript" file.
- Each figure must be uploaded as a separate file, and any multipanel figures must be assembled into one file.
- Manuscript: A .DOC version of the revised manuscript
- Figures: Editable, high-resolution, individual figure files are required at revision, TIFF or EPS files are preferred

For complete guidelines on revision requirements, please see the journal Submission and Review Process requirements at

<https://journals.asm.org/journal/Spectrum/submission-review-process>. **Submissions of a paper that does not conform to Microbiology Spectrum guidelines will delay acceptance of your manuscript. "**

Please return the manuscript within 60 days; if you cannot complete the modification within this time period, please contact me. If you do not wish to modify the manuscript and prefer to submit it to another journal, please notify me of your decision immediately so that the manuscript may be formally withdrawn from consideration by Microbiology Spectrum.

Response to Reviewers

We would like to thank the reviewers for their thoughtful comments and suggestions for the manuscript. We hope that our responses adequately address the comments, and the final revisions are acceptable. Our point-by-point responses to these comments are as follows:

Reviewer comments:

The IRB approval must be noted in the Methods section of the manuscript with the approval number and/or approval date.

Response: Thank you very much for your comments. This has been revised.

December 23, 2021

Dr. Zhuang guo
Hubei University of Arts and Science
No. 296 Longzhong Road Xiangyang, Hubei, China
Xiangyang
China

Re: Spectrum01442-21R2 (**Evaluation of bacterial diversity and evolutionary dynamics of gut *Bifidobacterium longum* isolates obtained from older individuals in Hubei Province, China**)

Dear Dr. Zhuang guo:

Your manuscript has been accepted, and I am forwarding it to the ASM Journals Department for publication. You will be notified when your proofs are ready to be viewed.

Sincerely,

Steven Frese
Editor, Microbiology Spectrum

Journals Department
Supplemental Material 4: Accept
Supplemental Material 2: Accept
Supplemental Material 3: Accept
Supplemental Material 1: Accept